# Aberrant CD3-Positive, CD8-Low, CD7-Negative Lymphocytes May Appear During Viral Infections and Mimic Peripheral T-Cell Lymphoma

**DOI:** 10.3390/diagnostics10040204

**Published:** 2020-04-07

**Authors:** Andreas Klameth, Andreas Neubauer, Christian Keller, Christian Aepinus, Ulrich Kaiser, Jörg Hoffmann, Cornelia Brendel

**Affiliations:** 1Department of Hematology, Oncology, Immunology, Philipps-University Marburg, Baldinger Strasse, 35043 Marburg, Germany; klametha@staff.uni-marburg.de (A.K.);; 2Department of Virology, Philipps-University Marburg, Baldingerstrasse, 35043 Marburg, Germany; 3Department of Hematology, Oncology and Palliative care, St. Bernward Hospital, Treibestraße 9, 31134 Hildesheim, Germany

**Keywords:** viral infection, T-cell lymphoma, aberrant CD3, positive CD8, low CD7, negative lymphocytes

## Abstract

Flow cytometry (FC) facilitates diagnosis of peripheral T-cell non-Hodgkin lymphoma (T-NHL), but overlapping features between reactive and neoplastic T-cell proliferations often hamper a rapid assessment. One hundred forty peripheral blood samples submitted to diagnostic FC for T-cell immunophenotyping were retrospectively analyzed. A T-cell population with a conspicuous aberrant surface epitope expression pattern was observed in 18 cases and diagnostic follow up was performed. The aberrant T-cell population exhibited a low scatter profile, a CD7-negative/low, CD8-low and CD3-positive immunophenotype, and monoclonal T-cell receptor expansion. T-NHL was ruled out by follow up in all cases. Epstein-Barr virus infection was diagnosed in 12 cases, cytomegalovirus infection in three cases; one patient had been vaccinated. The irregular subpopulation disappeared spontaneously within days or weeks. We describe a novel peripheral blood T-cell subpopulation with a low light scatter and CD8-low, CD7-negative/low and CD3-positive marker expression profile, which indicates reactive T-cell expansion in patients who present with peripheral lymphadenopathy and/or B symptoms.

## 1. Introduction

Flow cytometry (FC) immunophenotyping is a powerful diagnostic tool for characterization of lymphoproliferative disorders. T-cell neoplasms vary considerably with regard to immunophenotype and percentage within the total population of peripheral blood lymphocytes. Most T-cell non-Hodgkin lymphomas (T-NHL) lack exclusive surface markers that indicate a specific disease entity [1], and the overlap between reactive and neoplastic T-cell proliferation makes T-cell immunophenotyping a diagnostic challenge [2,3]. Particularly, Epstein–Barr virus (EBV)-associated T-cell lymphomas such as EBV-associated lymphoma of childhood may rapidly deteriorate and are often fatal due to multiorgan failure. FC-based fast diagnosis of peripheral T-cell lymphomas (PTCL) would therefore be desirable. Deletion, downregulation or aberrant expression of one or several T-cell surface markers or significant expansion of a single T-cell subpopulation are the main immunophenotypic criteria for the diagnosis of T-cell neoplasms. 

Analysis of the T-cell receptor (TCR) variable-ß-chain repertoire by FC may enhance diagnostic accuracy, but a panel of 24 fluorochrome antibodies covers about 70% of the normal human TCR-β-chain repertoire [4]. Besides, molecular analysis of TCR-ß-chain rearrangements is technically demanding and therefore limited to very few highly specialized diagnostic laboratories. Various conditions such as viral infections, inflammatory disorders or other reactive conditions can result in downregulation of the pan T-cell marker CD7 and thereby mimic T-cell neoplasms [5,6,7]. In particular, acute infectious mononucleosis (Epstein–Barr virus) can lead to an increase in cell size and thus forward scatter characteristics and variable changes in the expression of pan-T-cell markers [8].

Here we describe a novel T-cell surface marker pattern that mimics an aberrant T-cell population in 18 patients with viral infections such as Epstein–Barr virus or cytomegalovirus. Decreased cell size according to forward scatter profile and the unique pattern of lack of or weak expression of CD7 and normal/low expression of CD8 resolves after several days or weeks, without further treatment. Recognition and surveillance of this characteristic T-cell subpopulation therefore contributes to relieving the diagnostic dilemma because we can identify virus-induced T-cell alterations of proven benign origin.

## 2. Material and Methods 

### 2.1. Sample Specimens for Flow Cytometric Analysis

We reviewed our flow cytometry diagnostic reports with ambiguous results or suspected T-cell neoplasms over a period of seven years (from 2010 to 2017). The peripheral blood samples were collected freshly into ethylenediaminetetraacedic acid (EDTA) solution for in-house diagnosis or sent to our insitution for routine diagnostic purposes. According to the diagnosis suspected by the clinician, a B-cell- or T-cell-specific panel of antibodies was employed for immunophenotyping. Staining was delayed for one or two days when samples were sent from external hematologists. 

The study was performed according to the guidelines of the local ethics committee.

### 2.2. Panel for T-Cell Immunophenotyping

If T-cell lymphoma was suspected, an assay of seven tubes was employed. The antibodies were pipetted upfront into each tube. Tubes contained the following antibodies: tube 1 CD45/14 (BD Biosciences, San Jose, California USA, clone 2D1/ MφP9 FITC/PE) and CD19 (Invitrogen, Thermo Fisher Scientific, Waltham, Massachusetts, USA, clone SJ25-C1, APC); tube 2 CD4 (BD Biosciences, clone SK3, FITC), CD8 (BD Biosciences, clone SK1, PE) and CD3 (BD Biosciences, clone SK7, PerCP-Cy-5.5); tube 3 CD56 (BD Biosciences, clone NCAM 16.2, FITC), CD16 (BD Biosciences, clone B73.1, PE) and CD3 (BD Biosciences, clone SK7, PerCP-Cy-5.5); tube 4 CD2 (BD Biosciences, clone S5.2, FITC), CD5 (BD Biosciences, clone L17F12, PE) and CD3 (BD Biosciences, clone SK7, PerCP-Cy-5.5); tube 5 CD57 (BD Biosciences, clone HNK-1, FITC), CD11c (BD Biosciences, clone S-HCL-3, PE) and CD3 (BD Biosciences, clone SK7, PerCP-Cy-5.5); tube 6 TCR α/β (BD Biosciences, clone WT31, FITC), TCR γ/δ (BD Biosciences, clone 11F2, PE) and CD3 (BD Biosciences, clone SK7, PerCP-Cy-5.5); tube 7 CD7 (DAKO, Agilent, Santa Clara, California, USA, clone CBC.37, FITC), CD25 (DAKO, clone ACT-1, PE) and CD3 (BD Biosciences, clone SK7, PerCP-Cy-5.5).

A total of 100 μL of peripheral blood was added to each tube. After incubation for 15 min in the dark at room temperature (RT), FACS lysing solution (BD Biosciences) was added, followed by an additional five minutes of incubation. Then each tube was centrifuged for five minutes at 300× *g*. After washing with PBS and again centrifugation for five min at 300 g, samples were eluted in 500 μL FACS Flow (BD Biosciences) with 1% paraformaldehyde solution.

An IOTest Beta Mark PN IM3497 Kit (Beckman Coulter, Brea, California, USA) was used to test for clonal expansion of a specific V beta T-cell receptor, and staining was done according to the manufacturer’s instructions. Multiparameter flow cytometry was performed on a two-laser-equipped FACSCalibur (BD Biosciences) device or on a four-laser-equipped NAVIOS cytometer (Beckman Coulter, Version 1.3).

### 2.3. Panel for B-Cell Immunophenotyping

When exclusion of B-cell lymphoma was requested, two custom designed dry antibody reagent sample tubes, i.e., Duraclone^TM^ tubes (Beckman Coulter) were applied. Duraclone^TM^ one contained an antibody mixture of CD8/light chain kappa (clones B9, 11/RAHK, FITC), CD7/light chain lambda (clones 8H8/RAHL, PE), CD23 (clone 9P25, ECD), CD4/CD79b (clones 13B8.2/CB3.1, PC5.5), CD5 (clone BL1a, PC7), CD38 (clone LS198, APC), CD19 (clone J3-119, APC-A700), CD3/CD20 (clones UCHT1/HRC20, APC-A750), CD2/FMC7 (clones 39C1.5/ FMC7_F, PBE) and CD45 (clone J33, KrO). Duraclone^TM^ two contained a mixture of CD103 (clone 2G5, FITC), CD43 (clone DFT1.1.4, PE), CD25 (clone B1.49.907, ECD), CD10 (clone ALB1, PC5.5), CD200 (clone OX_104, PC7), CD11c (clone BU15, APC-A700), CD20 (clone HRC20, APC-A750), sIgM (clone SA-DA4, PBE) and CD19 (clone J3-119, KrO) antibodies.

From each sample, 100 μL of blood was taken and added to a 5 mL polystyrene tube. This was repeated to get two polystyrene tubes containing 100 μL of blood. Then 3 mL PBS buffer was added to each tube followed by centrifugation at 300 g for five minutes. The supernatant was removed. The washing step was repeated and the remaining liquid (about 100 μL) was resuspended by vortexing. Then a 50 μL cell suspension of each polystyrene tube was filled into Duraclone^TM^ one and two, respectively.

After fifteen minutes of incubation in darkness at RT, 2 mL red blood cell lysis buffer (Versalyse^TM^, Beckman Coulter) and 50 μL Fixative IO-Test3 (Beckman Coulter) were added to each Duraclone^TM^. After vortexing, the samples were stored in the dark for another fifteen minutes and subsequently centrifuged at 300× *g* and decanted. Again, 3 mL PBS was added to each tube and the tubes were then centrifuged at 300× *g* and subsequently decanted. A total of 500 μL PBS was added to each tube prior to analysis, which was performed on a NAVIOS cytometer (Beckman Coulter, Version 1.3).

For data analysis, we used Kaluza Analysis software, version 1.3 from Beckman Coulter and Microsoft Excel 2013. 

The average cell count per measurement and tube was 50,000 cells.

### 2.4. Cell Sorting with MoFlo

Cell sorting was performed on a MoFlow cytometer (DakoCytomation, Ft. Collins, CO, USA). For the sorting procedure, cells were gated with a combined live gate, according to scatter characteristics and dead cell exclusion via 4’,6-diamino-2-phenylindole (DAPI) stain at 100 ng/mL final concentration (Sigma, Taufkirchen, Germany). T-cells were discriminated via CD3 (BD Biosciences, San Jose, California USA, clone SK7, PerCP-Cy-5.5), CD4 (BD Biosciences, clone SK3, FITC), CD8 (BD Biosciences, clone SK1, PE) and CD45/14 antibody surface staining (BD Biosciences, clone 2D1/ MφP9 FITC/PE). B-cells were discriminated via CD19 (Invitrogen, Thermo Fisher Scientific, Waltham, Massachusetts, USA, clone SJ25-C1, APC). Sorting was performed in 1.5 mL polypropylene tubes at room temperature without external cooling. A total of 500,000 cells were sorted.

### 2.5. PCR Analysis for Virus Detection

We employed the Artus LC PCR Kit (Artus/Qiagen) as a ready-to-use system for real-time PCR analysis and detection of viral DNA with a LightCycler 2.0 (Roche) device. The “mastermix” contained reagents and enzymes for specific amplification of 97 bp of EBV DNA and 105 bp of cytomegalovirus (CMV) DNA. Additionally, the LC PCR Kit contained a second heterologous amplification system in order to detect potential PCR inhibition, which did not interfere with the detection limit of the analytic PCR. Analysis was done on a LightCycler 2.0 (Roche) machine. Quantification of viral DNA was performed using an external standard curve and a calibrator in each run.

## 3. Results

### 3.1. Patient Characteristics

We analyzed 140 peripheral blood samples that were sent for immunophenotyping because of suspected B- or T-cell lymphoma. In 18 (13%) patient samples, a unique aberrant antigen expression pattern (CD3-positive, CD8-low, CD7-negative/low) and low light scatter characteristics were found by FC. The 18 patients were eleven males and seven females, and patient age ranged from 15 to 77 years (mean 34 ± 22 years). Lymphoma was not confirmed in any of the 18 patients, either by FC or histology. Twelve patients had acute Epstein-Barr virus infection, diagnosed by a positive IgM antibody response and/or positive PCR testing of peripheral blood. Another three patients were diagnosed as having acute cytomegalovirus (CMV) infections, defined by a positive IgM antibody response and/or positive PCR testing of peripheral blood. In two patients, testing for viral infection was not performed but was suspected by the clinicians because of the corresponding symptoms and lack of a proven T-cell malignancy; thus, viral infection was suspected to be the most likely cause in all cases (Table 1). One patient (#17) had been vaccinated for measles, mumps and rubella with attenuated viruses. In all patients, symptoms resolved spontaneously during follow up. 

### 3.2. Flow Cytometry Detects a Lymphocyte Subpopulation with Low T-Cell Marker Expression

Immunophenotypic staining was performed in order to explore abnormal lymphocyte counts and/or cytology of atypical peripheral blood lymphocytes and/or suspected lymphoma due to symptoms or clinical findings. B- and/or T-cell lymphoma immunophenotyping was employed for diagnostic exploration. Eighteen patients had a distinct population of lymphocytes in common which appeared small (forward scatter (FSC)- and side scatter (SSC)-low) in FC analyses. 

All cells in the FSC/SSC-low region were predominantly T-cells with regular CD3 expression. Decreased expression of the T-cell marker CD8 or a separate CD8-positive population was observed in all 18 patients. Only CD7 downregulation was “almost always present” (16 out of 17); CD5 (5 out of 17) and especially CD2 (3 out of 17) were downregulated only in a few cases. CD4 was always negative. Some (9 out of 15) cases also showed diminished expression of TCR alpha/beta that was not associated with weak expression of CD3. The FC characteristics are summarized in Table 2 and displayed in Figure 1.

### 3.3. Analysis for T-cell Receptor Repertoire and Viral DNA

The samples of three patients were additionally analyzed for the T-cell receptor Vß repertoire of the conspicuous lymphocyte population with FC. In all three cases, monoclonal expansion of a specific T-cell receptor could be seen, as depicted in Figure 2. The Vß chains that were expressed in the three cases were Vβ-chain 18, Vβ-chain 5.1 and Vβ-chain 13.1. Thus, clonal T-cell expansion is not restricted to a certain Vβ-chain subtype. In contrast, an analysis for the T-cell receptor Vβ repertoire with no monoclonal T-cell expansion is provided in Appendix A. 

In order to determine whether the aberrant T-cell subpopulation was provoked by viral infection of CD8+ lymphocytes or perhaps was reactive in nature, we applied flow cytometry-based cell sorting (MoFlow, Beckman Coulter) to a patient sample (#6) with an aberrant T-cell population and with known EBV infection. After dead cell exclusion via DAPI stain, the T- and B-cells were separated for CD8-low, CD8-high, CD4+ (Figure 3), CD19+ and granulocytes. The subpopulations were analyzed for EBV-DNA by PCR (Table 3). EBV infection was detected in CD19+ B-cells and in the aberrant CD8-low cell population but not in the regular T-cell fraction of CD4+ or CD8-high cells or in granulocytes. 

### 3.4. Follow-Up Monitoring of the SSC/FSC-Low T-Cell Population

In five cases, follow-up samples were available. We observed that clearance of the irregular T-cells was rapid in some cases (3 out of 5) but prolonged in others (2 out of 5). However, spontaneous recovery occurred in every case. Quantification of aberrant T-cells as a percentage of total lymphocytes during follow up is depicted in Figure 4. In a median time of twelve and a half weeks (median 12.5 ± 28.5 weeks), the aberrant T-cell population disappeared and symptoms resolved in parallel.

## 4. Discussion

We describe a small aberrant CD8-weak/positive, CD7-negative/weak T-cell population that is associated with viral infections or with vaccination. The immunophenotype of this population had not been described before; therefore, T-cell non-Hodgkin lymphoma was ruled out by clinical work-up in every case. However, a clonal T-cell receptor rearrangement was detected in three samples. In all examined follow-up cases, the aberrant T-cell population disappeared and symptoms resolved in parallel. Cell sorting of a primary patient sample revealed that the aberrant CD8-positive T-cells were infected by EBV. Our data suggest that viral infections such as EBV or CMV, and in one case, immunization with attenuated measles virus, may provoke a clonal but benign expansion of a distinct subset of CD8-positive T-lymphocytes with a self-limited nature, which resolves completely during follow up within days or weeks. 

Viral infections often provoke fever or fatigue; especially, EBV infection is often associated with cervical lymphadenopathy, pharyngitis, splenomegaly and hepatomegaly. However, several lymphotropic virus infections are associated with T-cell lymphoma and sometimes such viral infections lead to rapidly deteriorating clinical courses, occasionally with fatal outcome, particularly in children and young adults [1,9]. Flow cytometry as a fast technique and a standard tool for the classification of peripheral lymphomas may therefore be useful in this scenario because T-cell lymphoma in particular is a potentially life-threatening disease with poor outcome [10]. Rapid FC diagnosis in order to distinguish T-NHL from reactive conditions is therefore an urgent clinical need. 

The differential diagnosis of aberrant lymphocytes in immune-compromised patients or in patients with a severe course of infection warrants rapid decisions by the treating physician but remains a diagnostic challenge as well.

In infectious mononucleosis, both B- and/or T-lymphocytes become activated. The vast majority of proliferating active T-cells are CD8+ T-cells reflected by a reduced CD4/CD8 ratio. Infected cells express human leucocyte antigen (HLA)-DR and CD28 but not CD11b, which may be related to the suppression of viral replication by mediating cytotoxic activity against infected B-cells [11,12,13]. In association with other viral infections such as cytomegalovirus (CMV) or human immunodeficiency virus (HIV), increased numbers of CD8+ cytotoxic suppressor T-cells have been reported [14,15], in line with our findings of an increased CD8 cell count resulting in a reduced CD4/CD8 ratio (Table 2).

Recent data provide evidence for coreactive CD8+ T-cells in hematologic diseases such as polycythemia vera [16], but with an immunophenotype different from the aberrant T-cell population described here. 

The mechanisms of how herpes virus-infected T-cells can eventually turn into malignant lymphoma cells is to our knowledge not completely understood. Association of viral clearance or transformation and human leukocyte antigen (HLA) may play a role in the malignant transformation of T- or NK cells [17]. Herpes viruses are equipped with immunoevasins, hereby facilitating the escape from antiviral immunity mechanisms [18]. Immunoevasins hamper antigen presentation via the major histocompatibility complex (MHC) I and II molecules [19].

Interestingly, vaccination with attenuated living viruses (MMR) also led to a distinct reactive T-cell population in one case. Therefore, it could be assumed that other viruses different from herpes virus may also provoke the unique T-cell expansion pattern described here. 

In summary, our findings suggest that viral infection such as EBV, CMV or the immunization with attenuated measles virus may provoke a clonal but benign expansion of a distinct subset of T-lymphocytes. Detection of T-cells with a distinct immunophenotype of low light scatter and a CD3-positive, CD7-negative/low and/or CD8-positive/low marker expression profile in patients with lymphadenopathy, night sweats and/or fever reflects a reactive self-limited T-cell expansion and should therefore be reported by diagnostic flow cytometry laboratories in order to provide supportive diagnostic information to physicians.

## Figures and Tables

**Figure 1 diagnostics-10-00204-f001:**
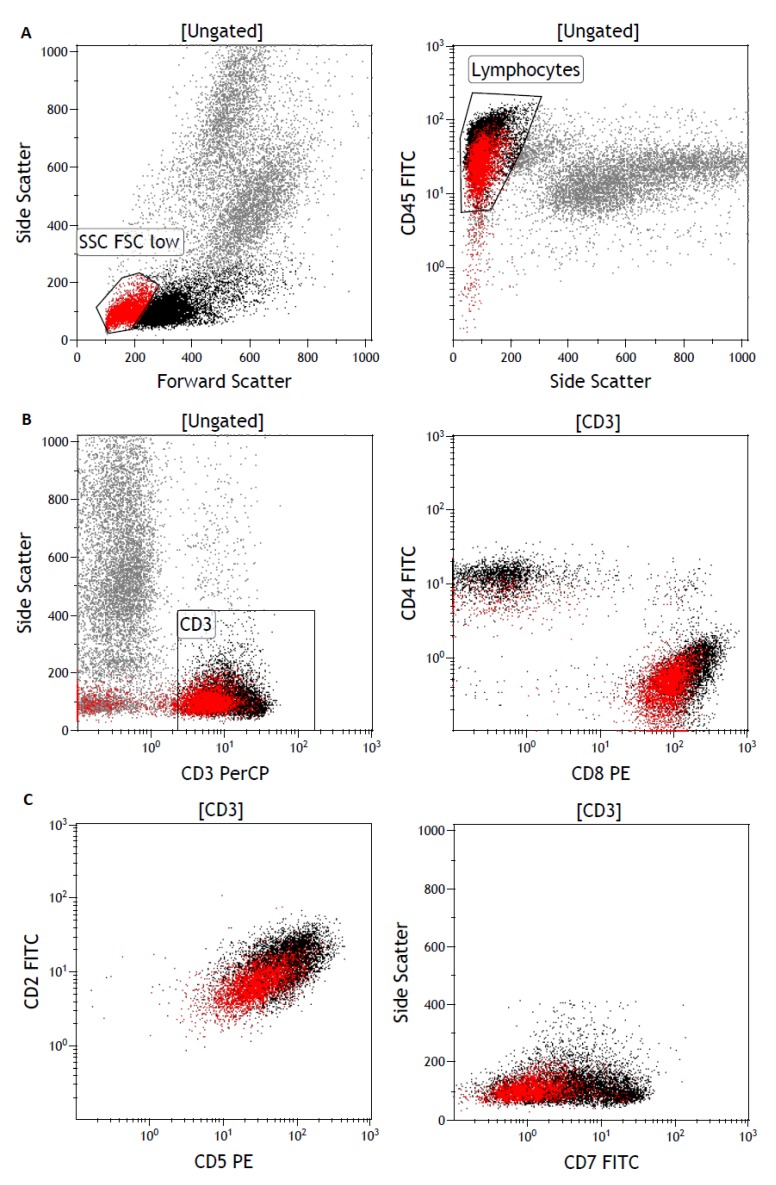
Gating strategy for the characteristic aberrant T-cell population. (**A**) Forward scatter versus side scatter gating identifies small aberrant cells indicated in red (left) with low CD45 expression (right). (**B**) The aberrant T-cells are CD3-positive (left). Plotting of CD4 versus CD8 highlights the lower expression of CD8 in these aberrant T-cells (right). (**C**) Diminished CD2 and CD5 marker expression on the small aberrant T-cell fraction (left); side scatter versus CD7 plotting demonstrates lack of/weak expression of CD7 (right).

**Figure 2 diagnostics-10-00204-f002:**
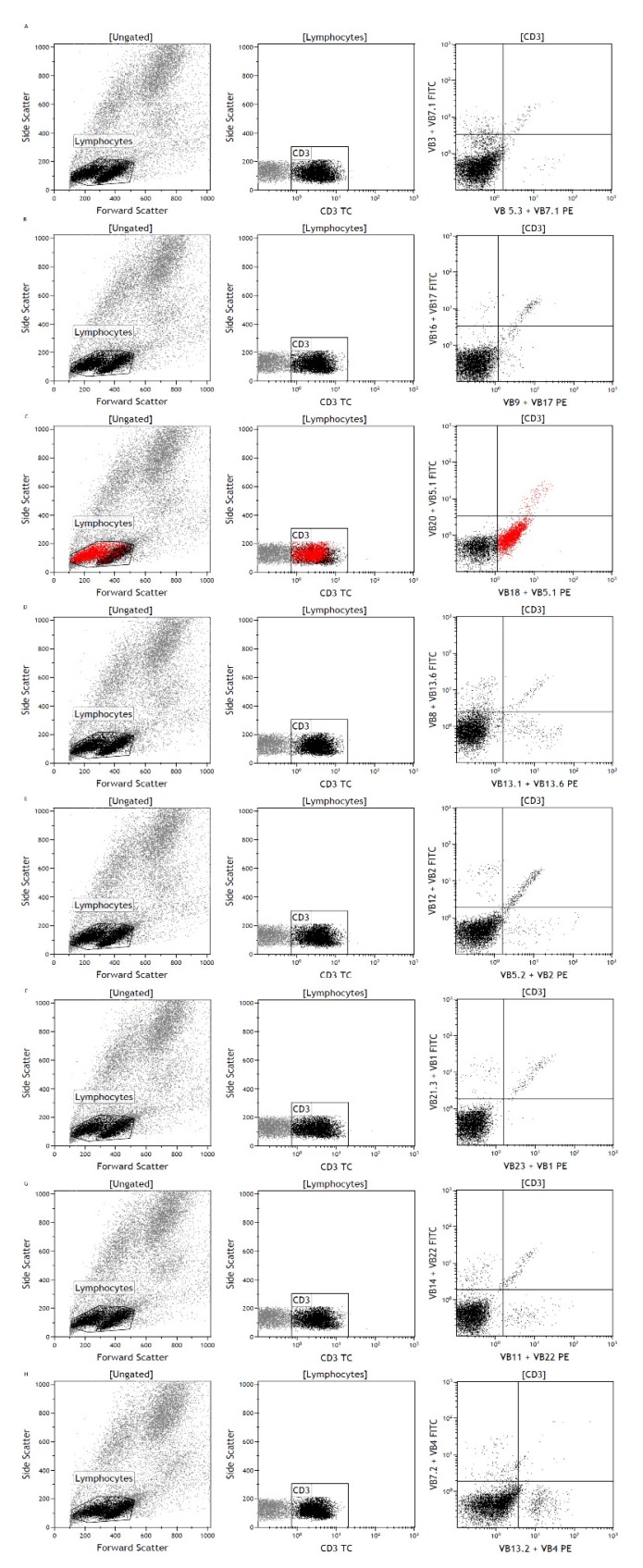
Clonal V beta T-cell receptor expansion on the aberrant cell population. In each row, gating on lymphocytes and CD3-positive cells (colored black) is shown (left and middle). On the right, analysis of Vß TCR of the CD3-positive cells is depicted. In row C, Vß TCR in Vß20/Vß5.1 versus Vß18/Vß5.1 is shown on the right. Monoclonal expression of Vß18 is clearly visible (colored red).

**Figure 3 diagnostics-10-00204-f003:**
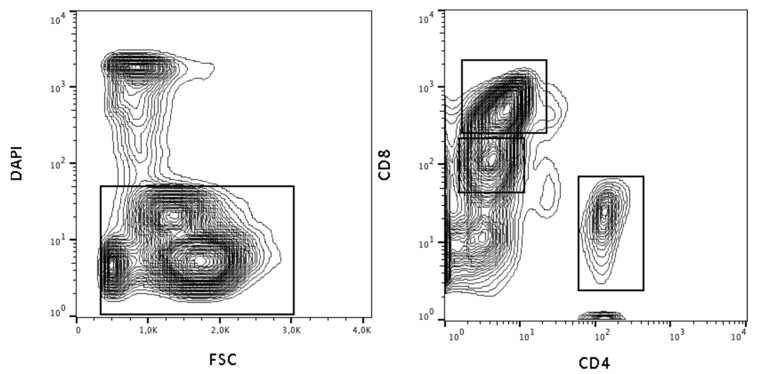
Cell sorting and dead cell exclusion. After dead cell exclusion via DAPI stain, the viable cells were further sorted and analyzed separately for viral infection. Dead cell exclusion is shown in the left plot. Sorting strategy for CD4, CD8-high and CD8-low cells is shown in the right plot.

**Figure 4 diagnostics-10-00204-f004:**
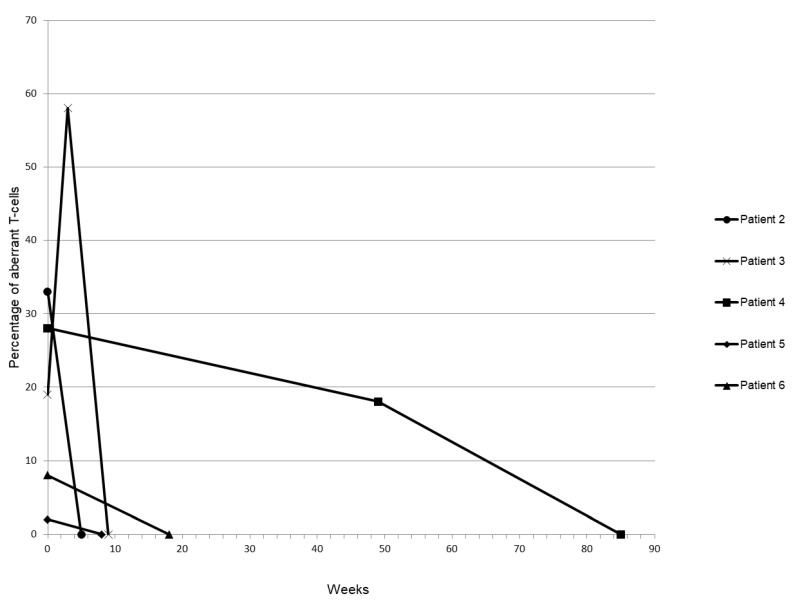
Time-course of aberrant T-cell clearance. The percentage of SSC/FSC-low aberrant T-cells per total lymphocyte count was determined throughout a follow-up period of several weeks in six patients.

**Table 1 diagnostics-10-00204-t001:** Patient characteristics.

Patient (no.)	Age (years)	Sex	% of WBC	% of Lymphocytes	Diagnosis
1	15	M	11	16	EBV
2	17	M	25	33	EBV
3	46	M	14	19	EBV
4*	77	M	17	28	Unknown viral infection
5*	31	M	24	31	EBV
6	65	M	4	8	Unknown viral infection
7	22	F	29	37	EBV
8	71	F	16	25	CMV
9	20	M	21	31	EBV
10	59	F	–	–	EBV
11*	60	F	20	37	EBV
12	59	M	4	5	CMV
13	26	F	14	31	EBV
14	45	F	14	22	CMV
15	18	M	2	2	EBV
16	17	M	7	12	EBV
17	37	F	10	24	MMR
18	37	M	1	1.3	EBV

Patient age, sex, percentage of aberrant SSC/FSC-low T-cells in all white blood cells (WBC), and percentage of aberrant T-cells in relation to all lymphocytes are depicted. * Indicates the patient samples on which TCR analysis was performed. Abbreviations: EBV, acute Epstein-Barr virus infection; CMV, Cytomegalovirus; MMR, Measles mumps rubella.

**Table 2 diagnostics-10-00204-t002:** Flow cytometric characteristics of the aberrant T-cell population. Scatter characteristics, expression of CD3, CD8, CD4, CD5, CD2, CD7, HLA-DR, T-cell receptor alpha and beta and CD4/CD8 ratio are depicted for each sample. “+++” indicates strong expression levels, “++” normal expression, “+” weak expression and “-” lack of expression. Abbreviations: n.i., Not investigated.

Patient (no.)	SSC/FSC	CD3	CD8	CD4	CD5	CD2	CD7	HLA-DR	TCR α/β	CD4/CD8
1	Low	++	++	-	++	++	-	+	++	0.07
2	Low	++	+	-	++	++	-	n.i.	+	0.18
3	Low	+	+	-	+	+	-	+	+	0.09
4	Low	++	+	-	+	++	-	n.i.	+	0.4
5	Low	++	++	-	++	++	-	n.i.	+	0.1
6	Low	++	+	-	++	++	-	n.i.	+	0.8
7	Low	++	++	-	n.i.	n.i.	n.i.	n.i.	n.i.	0.49
8	Low	++	++	-	++	++	-	n.i.	++	0.31
9	Low	++	++	-	+++	++	-	n.i.	+	0.21
10	Low	+++	++	-	++	++	-	n.i.	+++	1.8
11	Low	++	+	-	+	++	-	n.i.	+	0.32
12	Low	+++	++	-	++	+++	-	n.i.	++	0.17
13	Low	+++	++	-	++	++	-	+	+	0.18
14	Low	+++	++	-	++	++	-	n.i.	++	0.25
15	Low	+++	++	-	++	+++	+	n.i.	+++	0.04
16	Low	+++	++	-	++	+++	-	++	n.i.	0.11
17	Low	++	+	-	+	+	-	n.i.	+	0.28
18	Low	++	++	-	+	+	-	n.i.	n.i.	0.29

**Table 3 diagnostics-10-00204-t003:** EBV-PCR analysis of FACS-sorted cell subsets.

Population	CD8-High	CD4+	CD8-Low	CD19+	Granulocytes
EBV-PCR	negative	negative	positive	positive	negative

Patient T-cells were sorted for high and low expression of CD8 via flow cytometry. Additional sorting was performed for CD4+ T-cells, CD19+ cells and granulocytes. EBV PCR analysis was positive for T-cells with low CD8 expression but not for T-cells with a regular high CD8 expression level. CD19-expressing B-cells also gave a positive PCR signal for EBV.

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
