# Peer review of "Aberrant CD3-Positive, CD8-Low, CD7-Negative Lymphocytes May Appear During Viral Infections and Mimic Peripheral T-Cell Lymphoma"

_diagnostics, 2020, doi:10.3390/diagnostics10040204_

Round 1

Reviewer 1 Report

In this paper by klameth et al., authors report a novel peripheral blood T-cell subpopulation with a low light scatter and CD8low, CD7negative/low and CD3positive immunophenotype during viral infections (EBV and cytomegalo virus) and immunizations, using the rapid diagnostic flow cytometry approach. These immunophenotype mimics the peripheral T-cell lymphoma and therefore can aid the physician to do rapid diagnosis and avoid diagnostic dilemma of T-cell lymphoma.

The manuscript was composed well, and data support authors claims.

Minor suggestions;

  1. Figure numbers and legends-styling (bold letters) needs to be uniform. Figure 1 numbering appears to be repeated..authors should correct this…or introduce appropriate panel label if it is the continuation of same figure
  2. There are typos and grammar mistakes in several instances. Therefore, author should proofread well and correct these errors.

For eg..

Line 47--’Ebstein’ should be corrected as ‘Epstein’

Line 36 ‘EVB’ should be corrected as ‘EBV’

Line 152- depic-ted should be corrected as depicted

Line 175- posi-tive corrected as positive..

Line 195- ana-lysed--analysed

Reviewer 2 Report

Authors of the manuscript identified a novel, virus-induced T cell subpopulation characterized by a smaller cell size and CD7 negative/weak and CD8 weak/positive immunophenotype. These cells, present in 18% of patients with suspected lymphoma, were a result of benign clonal T cell expansion associated with viral infections or recent vaccinations and spontaneously resolved after a few days to weeks. The manuscript provides a quick, simple and potentially very useful (if confirmed by other labs and more patient samples) diagnostic method differentiating between T-cell Non-Hodgkin Lymphoma and benign and transient virus-induced lymphoproliferative disorder. I have just a few minor comments:
Line 36: “EVB-associated” should be EBV-associated
Line 53: “and commonly low expression of CD8” should be changed to:  “and normal/low expression of CD8” since low expression was observed only in 6 out of 18 samples (30%).
Table 1 needs some editing, empty rows between data for patients 14 and 15 should be removed, same the “Infektion” for patient 4.
Line 164: only CD7 downregulation was “almost always present”, CD5 (5 out of 17) and especially CD2 (3 out of 17) were downregulated only in a few cases.
Line 184: Which patient sample (Table 1) was analyzed for EBV-DNA? It is not clear what was the purpose of this experiment. The sorting strategy was different than the gating for the reported aberrant T cell subpopulation. Are the CD8 low/EBV-DNA positive cells the same as the low scatter, CD7 negative cells?
Finally, do authors have any biological explanation for the presence of this aberrant small T cell subpopulation? Is it possible that the smaller cell population represents cells undergoing apoptosis? The membrane of early apoptotic cells is intact (impermeable for live/dead dyes) but they shrink in size. Annexin V staining would solve this question.
